# In Silico Identification of lncRNAs Regulating Sperm Motility in the Turkey (*Meleagris gallopavo* L.)

**DOI:** 10.3390/ijms23147642

**Published:** 2022-07-11

**Authors:** Jan Pawel Jastrzebski, Aleksandra Lipka, Marta Majewska, Karol G. Makowczenko, Lukasz Paukszto, Joanna Bukowska, Slawomir Dorocki, Krzysztof Kozlowski, Mariola Slowinska

**Affiliations:** 1Department of Plants Physiology, Genetics and Biotechnology, Faculty of Biology and Biotechnology, University of Warmia and Mazury in Olsztyn, 10-719 Olsztyn, Poland; lukasz.paukszto@uwm.edu.pl; 2Department of Gynecology and Obstetrics, School of Medicine, Collegium Medicum, University of Warmia and Mazury in Olsztyn, 10-561 Olsztyn, Poland; aleksandra.lipka@uwm.edu.pl; 3Department of Human Physiology and Pathophysiology, School of Medicine, Collegium Medicum, University of Warmia and Mazury in Olsztyn, 10-561 Olsztyn, Poland; marta.majewska@uwm.edu.pl; 4Department of Animal Anatomy and Physiology, Faculty of Biology and Biotechnology, University of Warmia and Mazury in Olsztyn, 10-719 Olsztyn, Poland; karol.makowczenko@uwm.edu.pl; 5Department of Botany and Nature Protection, Faculty of Biology and Biotechnology, University of Warmia and Mazury in Olsztyn, 10-719 Olsztyn, Poland; 6In Vitro and Cell Biotechnology Laboratory, Institute of Animal Reproduction and Food Research, Polish Academy of Sciences in Olsztyn, 10-748 Olsztyn, Poland; j.bukowska@pan.olsztyn.pl; 7Department of Entrepreneurship and Spatial Management, Institute of Geography, Pedagogical University of Cracow, 30-084 Kraków, Poland; sdorocki@up.krakow.pl; 8Department of Poultry Science, Faculty of Animal Bioengineering, University of Warmia and Mazury in Olsztyn, Oczapowskiego 5, 10-719 Olsztyn, Poland; kristof@uwm.edu.pl; 9Gamete and Embryo Biology, Institute of Animal Reproduction and Food Research, Polish Academy of Sciences in Olsztyn, 10-748 Olsztyn, Poland; m.slowinska@pan.olsztyn.pl

**Keywords:** lncRNA, RNA-seq, NGS, DEL, gene expression, sperm motility, transcriptomics, turkey

## Abstract

Long non-coding RNAs (lncRNAs) are transcripts not translated into proteins with a length of more than 200 bp. LncRNAs are considered an important factor in the regulation of countless biological processes, mainly through the regulation of gene expression and interactions with proteins. However, the detailed mechanism of interaction as well as functions of lncRNAs are still unclear and therefore constitute a serious research challenge. In this study, for the first time, potential mechanisms of lncRNA regulation of processes related to sperm motility in turkey were investigated and described. Customized bioinformatics analysis was used to detect and identify lncRNAs, and their correlations with differentially expressed genes and proteins were also investigated. Results revealed the expression of 863 new/unknown lncRNAs in ductus deferens, testes and epididymis of turkeys. Moreover, potential relationships of the lncRNAs with the coding mRNAs and their products were identified in turkey reproductive tissues. The results obtained from the OMICS study may be useful in describing and characterizing the way that lncRNAs regulate genes and proteins as well as signaling pathways related to sperm motility.

## 1. Introduction

The formation and maturation of the sperm in testis, known as spermatogenesis, is a series of complex molecular processes and morphological changes, including mitotic and meiotic divisions and differentiation leading from germ cells to mature spermatozoa [1,2]. These processes are under multiple regulators, such as hormones [3], genetic factors [4,5], long non-coding RNA (lncRNA) and epigenetics [6,7].

The molecular mechanism of sperm production from the primordial germ cells is well-documented in mammals, but there are only a few studies detailing this process in birds [5,8]. The final product of spermatogenesis is a spermatozoon that consists of the head, which is the carrier of the male genetic material, and the flagellum, which provides the motive force to move towards the oocyte. Spermatogenesis consists of several stages, including spermatocytogenesis, mitosis, meiosis and spermiogenesis [9]. Spermiogenesis is the transformation of spermatids into spermatozoa without further cell division. It includes the formation of an acrosome and an axoneme, loss of cytoplasm, and the replacement of nucleohistones with nucleoprotamine, which accompanies nuclei condensation [10,11]. In avians, spermiogenesis varies from species to species, e.g., in guineafowls, spermiogenesis involves ten distinct morphological steps [12]. However, in red junglefowl, spermiogenesis entails 8–10 steps [13,14,15] and 12 stages in Japanese quail [16,17].

The only function of the sperm is to fertilize the egg. For this, the sperm must be highly mobile, especially in the female reproductive system. Since sperm is transcriptionally inactive, mobility depends on flagellum and actin element formation in testes and post-testicular development of sperm motility [18]. Mobility dysfunction, which may be impaired at the molecular level, handicaps sperm and leads to increased infertility up to the complete loss of fertilization possibility [19,20].

The turkey (*Meleagris gallopavo*) is one of the most important livestock animals and the second largest contributor to the world’s poultry meat production [21]. The industrial turkey breeding is based on the artificial insemination because natural mating results in adequate fertility levels [22]. In comparison to other Galliformes species, turkey semen is characterized by some unique features, such as oxidative respiration as the main pathway in spermatozoa energy metabolism [23,24,25], distinctive proteome pattern of seminal plasma [26,27] and low semen sensitivity for cooling/freezing procedures [28]. The molecular mechanisms and key factors responsible for producing spermatozoa with good quality are still poorly understand for turkey spermatogenesis. So far, only the pathways associated with specificity of the reproductive tract have been described for turkey [5]. There is a lack of information describing molecular mechanism of spermatogenesis such as lncRNAs that can be involved in regulation of mature spermatozoa quality.

Semen quality is of economic importance and is studied to optimize reproductive performance [29] and scientific significance [30]. Investigating the genetic and epigenetic mechanisms underlying spermatogenesis allows us to better understand the entire process of sperm production and maturation on the molecular level. Sperm motility is the basic parameter of semen quality [4,31,32,33] in both mammals and birds [34], including turkeys [35].

Semen quality, as well as sperm motility, is influenced by gene expression and regulatory factors such as lncRNAs [7]. LncRNA is the most abundant group of transcripts apart from coding sequences and may be crucial in regulating gene expression and protein activity in maintaining basic cell functions as well as spermatogenesis [6,36]. In the current research, we focused on the identification of lncRNAs in the whole transcriptome of three turkey tissues: testis (T), epididymis (E), and ductus deferens (DD). Moreover, the deep in silico analyses revealed the potential interactions between differentially expressed lncRNAs (DELs) and differentially expressed genes (DEGs) engaged in the sperm motility within the male reproductive tract tissues.

## 2. Results

### 2.1. Sequencing, Mapping and Expression

As the result of sequencing, more than 100 million reads were obtained in each sample (Table 1). The highest number of raw reads was obtained for one of the individual T samples (165.8 raw reads), and the lowest was for the DD sample (102.6 of raw reads). The highest value of sequencing depth for both individual samples and mean was in the epididymis samples (176.9 million of reads in individual samples, mean = 152 million of reads). The trimming process reduced the quantity by about 15%, resulting in mean values of 83.22%, 83.47% and 84.33% of processed reads in ductus deferens, epididymis and testes, respectively. The mapping efficiency ranged from 96.6% to 98.9% with means of more than 98% in each group of samples. The count calling process resulted in the expression values for 56,037 genes and 103,847 transcripts in all samples. The number of expressed genes in individual samples ranged from 17,596 (“12_E”) to 43,510 (“14_T”), while the number of expressed genes in three analyzed tissues was 53,385 in testes, 35,313 in ductus deferens and 41,029 in epididymis. The number of expressed transcripts ranged from 22,109 (“12_E”) to 73,569 (“14_T”), and in summary for each tissue: testes—94,726, ductus deferens—64,905 and epididymis—75,765.

### 2.2. lncRNA Identification

The set of all pooled transcripts (103,847) from all samples was filtered following the pipeline presented in Figure 1. The whole assembled transcriptome contained 55,168 sequences of more than one exon, 86,326 sequences longer than 200 nucleotides, 87,374 non-protein coding and 100,248 transcripts with an expression of more than ten counts (Figure 2A).

The dataset of filtered sequences was then analyzed for coding potential using seven different tools. The results of various methods were different. The highest number of non-coding sequences was obtained from CPC2—20,344, PLEK—20,168 and CNCI—17,931, whereas FEELnc, CPAT, LncFinder and MDeep resulted in 1262, 1085, 922 and 2582 non-coding sequences, respectively (Figure 2B). Due to such differences, it was decided to apply the consensus that, if a given sequence, was confirmed by at least five methods (out of seven), it was qualified as non-coding. As a final result, 4391 lncRNA transcripts were obtained, where 863 were predicted and 3487 were annotated using the reference turkey ENSEMBL genome (version 5.01.104).

### 2.3. Differentially Expressed lncRNAs

The final lncRNA dataset was used to identify the differentially expressed lncRNAs (DELs) in each comparison. It resulted in 139 DELs in “epididymis/testis” comparison (93 up- and 46 downregulated), 182 DELs in “ductus deferens/testis” (132 up- and 50 downregulated) and 25 only upregulated DELs in “epididymis/ductus deferens” comparison (Figure 3, Table 2 and Appendix A).

Potential interactions between all identified DEGs and DELs within each comparison were analyzed (Appendix A), but only relations with DEGs involved in sperm motility processes, described in detail by Słowińska et al. [5], were further visualized, described and discussed (Table 3, Figure 4). Several CIS-relations were detected between DEGs and DELs (Table 4), but no such relation was found for genes involved in sperm motility. This is why these results were presented as Appendix A and are not discussed in the manuscript.

**Figure 4 ijms-23-07642-f004:**
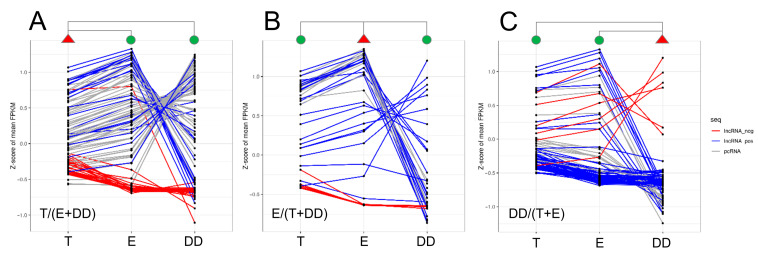
The expression profiles of DELs in three comparisons based on trans-correlated DEGs: (**A**) DEGs of statistically significant expression difference between ductus deferens (DD) and other tissues-testis (T) and epididymis (E), (**B**) epididymis vs. testis and ductus deferens, (**C**) testis vs. epididymis and ductus deferens. The red triangle indicates the tissue against which the expression profiles of the other two tissues (green circles) were compared. The expression profiles of DEGs are presented as grey lines, blue lines are the profiles of positively correlated DELs and red are negatively correlated lncRNAs. Expression profiles (the Z-score values of FPKM-Fragments Per Kilo base of transcript per Million mapped fragments) of all identified DELs are presented as the heatmap in Figure 5.

**Figure 5 ijms-23-07642-f005:**
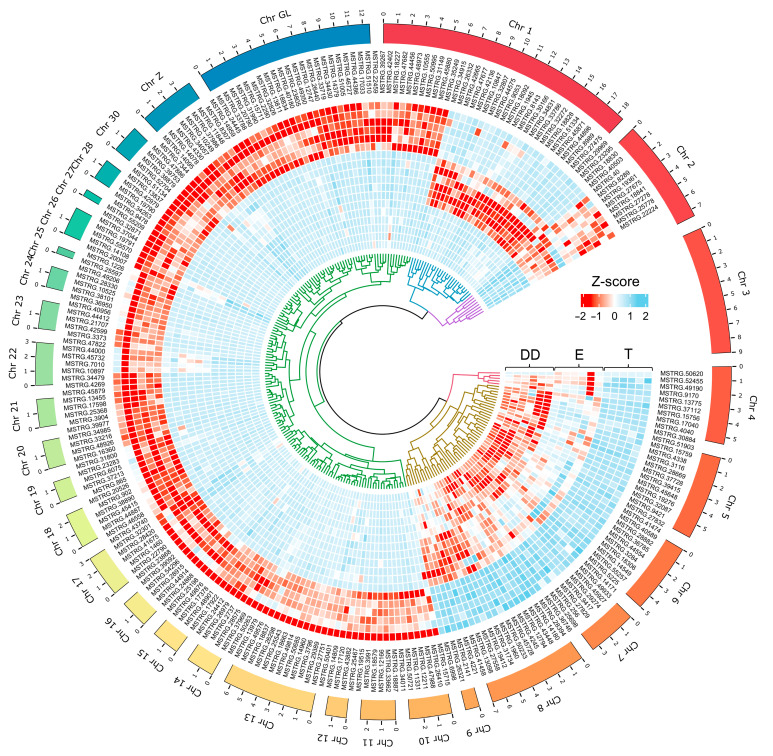
The expression heatmap (the Z-score of the FPKM value) of the identified lncRNAs in each sample of each tissue grouped in the clusters (the dendrogram inside). The outer bars’ length corresponds to the chromosomal distribution of the DELs.

### 2.4. TRANS-Acting Genes

Analyses of the correlation of expression profiles resulted in 33,315 unique relations between DEGs and DELs. By limiting the results to the relationships between DELs and genes associated with the processes responsible for sperm motility 519 (32 DELs with 78 DEGs), 749 (72 DELs with 85 DEGs) and 99 (7 DELs with 28 DEGs) correlations were detected, respectively, in “epididymis/testis”, “ductus deferens/testis” and “epididymis/ductus deferens” comparison (Table 4). Detailed information about the types of relationships between individual protein-coding genes and DELs is presented in Appendix A, and the summary of these relations for genes associated with sperm motility is available in Appendix A.

The number of trans-correlations between DELs and DEGs involved in the DD/(T + E) group of sperm motility processes (the processes enriched for genes of significantly different expression between ductus deferens and both other tissues) was 229 (blue links—positive correlations and red—negative correlations in Appendix A). Fourteen DEGs (the middle column in Figure 6 and grey lines in Figure 4A) were associated with six processes (microtubule cytoskeleton organization, microtubule-based movement, flagellated sperm motility, axonemal dynein complex assembly, outer and inner dynein arm assembly) showing statistically significant co-expression with 70 DELs (blue and red lines in Figure 4A and the third column in Appendix A). Eleven genes of significantly different expression between the epididymis and other two tissues (group E/(T + DD)), associated with three processes (calcium ion binding, lipid metabolism and reproductive process), have significant co-expression with 29 DELs (Figure 4B and Figure 6A) by 88 relations. Genes that differ in expression between testis and other analyzed tissues (55 genes) were engaged in 12 processes related to sperm motility (six actin building processes, protein phosphorylation, apoptosis, response to stress, estrogen and oxygen-containing compounds and cation symporter activity) and showed 644 co-expression relations with 48 DELs (Figure 4C and Appendix A). Almost all genes within each process showed a high and statistically significant correlation of expression with DELs. There were only a few genes without such co-expression. This is why the knowledge of each relationship should be extensively studied and enriched by additional analyses.

### 2.5. Direct Interactions

A total of 3101 RNA-RNA interactions were identified between 124 transcripts (50 DEGs) and 188 lncRNAs in the T/(E + DD) group (Appendix A). Moreover, 796 potential interactions between 25 transcripts of 10 DEGs and 173 lncRNAs in the group E/(T + DD) (Figure 6B) were discovered. A total of 607 unique interactions between 26 transcripts of 14 DEGs and 163 lncRNAs were distinguished in the DD/(E + T) group (Appendix A). The direct RNA–protein interactions’ analyses resulted in 5951 highly probable relations between 42 proteins (20 DEGs) and 338 lncRNAs in the T/(E + DD) group, 1089 results for 255 lncRNAs and nine targets (4 DEG) lncRNAs in the group E/(T + DD) and 757 interactions between 154 lncRNAs and eight protein sequences (3 DEGs). All detailed data on the interactions are stored in Appendix A. The genes that showed the potential for RNA-RNA interactions (and co-expressing) but did not interact in protein type-RNA with DELs in the group DD/(T + E) were: *DNAH7* and *ZMYND10*; in group E/(T + DD), these genes included *DNAH5*, *CDHR2*, *CDH20;* and, in the T/(E + DD) group, these genes were: *SDCBP*, *CTB1*, *FAT1*, *TWF1*, *ACVR1*, *ARPC5*, *NEDD9*, *MET*, *SORBS1*, *ATP1A1*, *KRT19*. The genes that showed the potential for RNA-protein but not RNA–RNA interactions in the DD/(T + E) group were: *PLK4* and *LRRC6* in group E/(T + DD) are *CLGN,* and in the group T/(E + DD) were: *PACSIN2* and *POF1B*. The *CCDC40* gene was detected in all analyzed types of correlations and interactions in the group DD/(T + E), *LRP2* in group E/(T + DD), and the genes that showed all co-expression, RNA–RNA and RNA–protein interactions were: *SLC9A3R1*, *SLIT2*, *GATA3*, *NET1*, *GSN*, *SPRY2*, *PDLIM3*, *BAG3*, *EVL*, *APP* and *EZR*. The analyses of functional annotations using STRING resulted in 27 nodes and 21 edges with a *p*-value equal to 1.41 × 10^−5^ (see Figure 7).

### 2.6. Real-Time PCR-Validation of the Expression of Selected lncRNAs

The real-time PCR analysis, in all cases, confirmed the NGS expression levels of selected lncRNAs in turkey testis, epididymis and ductus deferens (Figure 8).

## 3. Discussion

In this study, lncRNAs were identified in the transcriptome of three turkey tissues, i.e., testes, epididymis and ductus deferens. In particular, the study focused on the analysis of lncRNAs, which can regulate the activity of genes and proteins involved in sperm motility. The present studies are a continuation of the Slowińska et al. [5] study. However, the current studies are the first attempt to identify and analyze the differences in the expression of lncRNAs that may have the potential to regulate sperm motility.

Among all 4391 identified lncRNAs, 139, 182 and 25 DELs were discovered in the following cases: “epididymis/testis”, “ductus deferens/testis” and “epididymis/ductus deferens”, respectively. The study analyzed the distance relations (CIS), co-expression (TRANS), RNA–RNA and protein–RNA direct interactions between DELs and DEGs for each comparison. Based on these results, lncRNAs were indicated which may regulate processes related to sperm motility. The results obtained from the OMICS study may be useful to describe and characterize the way that lncRNAs regulate genes and proteins as well as signaling pathways related to sperm motility.

### 3.1. The Regulations between Testes and Post-Testicular Areas of Spermatogenesis

#### Testes (Group T/(E + DD))

The evidence indicates that testicular and post-testicular development of sperm varies substantially [37]. Consistent with previous results [5], the differences in the expression of DELs between studied tissues were observed. The abundance and diversity of the revealed DELs with strong and statistically significant correlations as well as potential interactions suggest that lncRNAs may play an important role in the genetic background by determining differences in processes associated with sperm maturation and motility, as well as the entire spermatogenesis. The genes that had significant differences in expression between the testis and epididymis and ductus deferens revealed the potential for interaction with lncRNA and thus the ability to regulate spermatogenesis processes in turkey. The current study identified lncRNAs that may affect mentioned genes involved in the spermatogenesis, in particular those related to sperm motility. One such gene whose activity is differentially regulated between the testes and the ductus deferens that may be crucial in sperm motility is *ARPC5* (short names of genes and macromolecules are explained in the Abbreviations section). ARPC5 takes part in actin polymerization, essential for sperm motility and capacitation processes. Moreover, it acts through the actin nucleation for the production of the actin filaments. The ARP2/3 complex consists of the two proteins ARP2 and ARP3 and five additional subunits ARPC1-5 [38,39]. ARP2 and ARP3 are the core of the mechanism that catalyzes the nucleation of actin fibers by ATP hydrolysis [40,41,42], while the remaining subunits stabilize the ARP2/3 complex [43]. The ARP2/3 complex plays a pivotal role in the regulation of sperm function and male fertility through actin polymerization [44]. The authors’ research of both co-expression analyses and direct RNA–RNA interactions revealed that the *ARPC5* gene may be regulated by numerous lncRNAs. Both positive and negative co-expression was confirmed in the testis vs. epididymal (Appendix A) and testis vs. ductus deferens comparisons (Appendix A). However, in the case of direct interactions, a potential interplay was demonstrated only at the RNA-RNA level, and *ARPC5* has a high probability of interacting with more than 60 DELs, and 18 of those DELs have the potential to interact only with the *ARPC5* (Appendix A).

The current results identified two lncRNAs (*MSTRG.13455* and *MSTRG.34057*) that had a negative correlation with the expression of *ARPC5*. This phenomenon indicated that the expression of *MSTRG.13455* and *MSTRG.34057* in the testicles was higher than in other tissues. In contrast, the direct interaction of those lncRNAs has been demonstrated with other genes related to sperm motility, i.e., *SPTBN1*, *NEDD9*, *CGNL1* and *MYH11*.

SPTBN1 connects the cell membrane with the actin cytoskeleton and plays a role in determining the shape of the cells, transmembrane protein distribution and organization of organelles [45]. The current research indicates that the SPTBN1 protein can directly interact with a few identified lncRNAs (*MSTRG.13455*, *MSTRG.34057*, *MSTRG.35249*, *MSTRG.45880*). The expression of the aforementioned lncRNAs was elevated in the testes, but the RNA–protein interaction may take place in the area of further stages of spermatogenesis, which is suggested by the negative correlation of expression. It was also shown that *MSTRG.13455*, *MSTRG.31149*, *MSTRG.35249* lncRNAs can directly interact with CGLN1 protein. The physiological expression profile of *CGNL1* plays a key role in testes development through the differentiation and dynamics of the Sertoli cell cytoskeleton. Moreover, non-disturbed expression and full activity of the CGLN1 are required for the appropriate flow of spermatogenesis [46]. In turn, NEDD9 regulates actin dynamics through cortactin deacetylation [47].

In this research, it was hypothesized that *LEF1* may be regulated by lncRNA on both RNA–RNA and RNA–protein levels. The *LEF1* is involved in the apoptotic process and is an important member of the Wnt/β-catenin signaling pathway. Moreover, it was reported that *LEF1* may contribute to the arrest of spermatogonial differentiation [48]. *ANXA1* plays diversified roles in cell functioning. It affects cell proliferation and differentiation as well as plasma membrane repair and apoptosis [49]. The expression level of *ANXA1* is related to semen DNA fragmentation, making it a possible new biomarker for sperm DNA quality [50]. Both *NRP1* and its cooperation with *VEGFA* are critical for vascular development in various angiogenic processes. It is known that *NRP1* enhances signal transduction of *VEGFA* isoforms. It is suggested that decreased expression of the *NRP1* in Sertoli cells may inactivate *VEGFA* and thus ultimately hinder the establishment and proliferation of the spermatogonial stem cells (SSCs), leading to an imbalance between undifferentiated spermatogonia self-renewal, differentiation and impaired spermatogenesis [51]. The SSCs may also be regulated by *ITGB1* as an indirect way of response to androgens that involves interaction with E-cadherin on SSCs to regulate its fates [52,53]. Furthermore, *SORBS1* is associated with the proper function of microtubules and cytoskeleton structure. Moreover, it can also be involved in the formation of multinuclear round spermatids and the formation of mature sperm [54]. Stöckl et al. revealed that *CNN3* regulates the actin cytoskeleton and is involved in cell fusion and myogenesis. However, its specific function in spermatogenesis is still unknown [55]. Nevertheless, it may be speculated that *CNN3* may be engaged in cytoskeletal dynamics and the dramatic transformation that the SSCs undergo to ultimately produce highly specialized spermatozoa [56]. *NET1* is also involved in the control of the cell cycle, and through regulation of rearrangements of actin, cytoskeleton can control cell motility [57,58]. The methylation and expression level of *GATA3* can affect sperm motility and male fertility status [59]. *MICAL2* covalently modifies and affects actin [60] and, due to its interactions with cas proteins, may regulate integrin-mediated cell adhesion and cytoskeletal organization [61].

Differentially expressed lncRNAs detected between analyzed turkey male reproductive tissues have the potential to regulate the aforementioned genes. Moreover, the current results indicated that lncRNAs may have the potential for multilevel regulation of actin and cytoskeleton modifications, as well as basic cell functions including proliferation and differentiation. Furthermore, these processes and functions are known to be crucial for appropriate sperm production; however, the current research gives an in-depth picture of how they may be regulated.

### 3.2. Post-Testicular Development

#### 3.2.1. Epididymis (Group E/(T + DD))

In the epididymis, among all DELs, five lncRNAs were identified with significantly different expressions than in the other two analyzed tissues. Those lncRNAs showed a high potential for direct RNA–protein interactions with multiple proteins. The results indicate that *MSTRG.45728*, *MSTRG.44914* and *MSTRG.39415* may have the ability to directly regulate the activity of the PLA2G12B and CLGN proteins. Moreover, *MSTRG.18092* is likely to interact with PLA2G12B, CLGN, HNF4A and LRP2. The HNF4A is a member of the nuclear transcription factors and contains a conserved nucleic-acid binding domain (Figure 9) [62].

Interestingly, *HNF4A*, *CLGN* and *LRP2* genes can be regulated by lncRNAs at the level of RNA–RNA interactions. Moreover, another two lncRNAs were identified, i.e., *MSTRG.45871* and *MSTRG.47677*, whose expression profiles revealed a high correlation (r > 0.9, *p*-value < 1 × 10^−4^) with the *LRP2*, as well as a high probability of direct interaction with its product. Possible direct interactions were also identified on the RNA–RNA level with genes such as *DNAH5*, *HEXB*, *CDHR2*, *CDH20*, *HPGDS* and *RDH10*. The *DNAH5* gene product is involved in the assembly of the outer axon dynein arm of flagella or cilia, and a mutation within this gene contributes to a risk of male infertility [63,64]. It is highly possible that any changes in its expression may affect post-testicular sperm maturation. According to available data, *HPGDS* may play a key role in the regulation of the spermatogenesis mechanism [65]. On the other hand, the *HEXB* gene is involved in the binding of the sperm to zona pellucida [45,66], which is crucial for effective fertilization. However, there is no information on those genes’ function in the epididymis. However, taking into consideration that epididymis is important for avian sperm to achieve appropriate quality for fertilizing capacity and motility [5,67], it can be expected that presently identified lncRNAs with multivariate interactions on different levels may have huge potential to regulate the course of the spermatogenesis and sperm maturation.

#### 3.2.2. Ductus Deferens (Group DD/(T + E))

Detailed analysis showed that the processes related to the formation of dynein arms can be regulated by the cross-talk between lncRNA with the following genes: *ARMC4*, *CCDC40*, *DNAH7*, *LRRC6*, *PIH1D3*, *ZMYND10*, *DEUP1*, *NDC80*, *PLK4*, *RSPH9*, *TTC26*, *CFAP221*, *ROPN1L* and *CFAP69*. DEGs associated with inner and outer dynein arm assembly in epididymis were also detected and described in the authors’ previous study [5]. The high value of the co-expression correlation coefficient between *CCDC40* and lncRNA, which was identified both in the analyses of direct RNA–protein interactions and in the analysis of expression profiles: *MSTRG.17922* (r = 0.97 *p*-value = 1.88 × 10^−7^) and *CCDC40* and *MSTRG.45871* (r = 0.92, *p*-value = 2.18 × 10^−5^), suggest that these two lncRNAs may play a supporting role in the formation of protein complexes of dynein arms (*MSTRG.17922* log_2_(FC) = 2.35 and *MSTRG.45871* log_2_(FC) = 2.65 in comparison DDE). The *ARMC4* gene shows a high co-expression correlation (r = 0.94, *p*-value = 6.26 × 10^−6^) with *MSTRG.1945* lncRNA. This lncRNA is highly overexpressed in the epididymis in relation to the ductus deferens (log_2_(FC) = 10.2). The *DNAH7* reflects the potential for direct interaction and co-expression with *MSTRG.47677* lncRNA, which is expressed in the epididymis but not identified in the ductus deferens. It has also been revealed that two lncRNAs characterized by high overexpression in the testes relative to the ductus deferens (*MSTRG.34479*: log_2_(FC) = 3.59 and *MSTRG.40956*: log_2_(FC) − 6.39) depicted very similar expression profile with *DNAH7* (in both cases r = 1 and *p*-value = 9.46 × 10^−7^ and 7.45 × 10^−9^ respectively) and a high potential for direct RNA–RNA interaction. *ARMC4* may have an evolutionarily conserved function in spermatogenesis by being involved in cilia/flagella assembly, structure or function [68,69], and DNAH7 is a component of the inner dynein arm of ciliary axonemes [70]. Interestingly, the *CCDC40* plays a key role in cilia formation and function and is an element of the dynein regulatory complex [71]. Mutations within the *CCDC40* are associated with cilia and sperm dysmotility [72], and are suspected to be involved in idiopathic male infertility [64]. Moreover, co-expression and interactions between lncRNAs and the *CCDC40* and *ZMYND10* are involved in bovine endometritis [73]. These distant examples demonstrate that those genes are prone to the regulatory function of the lncRNAs. Hence, the proper formation of the cilia may be one of the processes regulated by such an interaction.

All DELs detected in the current study were novel lncRNAs, not yet annotated; therefore, their potential function was discussed based on postulated functions of genes with which they interact or correlate. However, it should be mentioned that the impact of lncRNAs not only manifests through gene expression regulation. Besides transcriptional level regulation, at the post-transcriptional level, lncRNAs may be involved in mRNA processing, as well as direct interactions with proteins to regulate translation and post-translation modifications [74,75]. Within the authors’ research, numerous lncRNAs interactions as distance relations (CIS), co-expression (TRANS), RNA–RNA and protein–RNA direct interactions were detected. CIS-regulation can act by two main, independent mechanisms that do not exclude each other, the lncRNA can regulate neighboring loci (within the same chromosome), and/or can affect chromatin state or steric impediment that influences the expression of nearby genes [76]. Conversely, lncRNAs act as *trans* when they affect genes located on other chromosomes [36]. RNA–RNA interactions may also refer to interactions between two RNAs (inter-molecular) or between different regions of one RNA molecule (intra-molecular) and both mechanisms are significant for RNA function and regulation [77]. RNA–protein interactions act through the formation of dynamic ribonucleoprotein complexes [78]. Although the nature and dynamics of the aforementioned interactions still need to be investigated [79], nevertheless, multiple features of lncRNAs can determine their functionality. These features include sequence, expression level, processing, cellular localization, structural organization and interactions with other molecules. Moreover, the regulatory effect is mediated by the lncRNA transcript or by its transcription [75,79].

The current research focused on the comparison of gene and lncRNA expression in different reproductive tissues regardless of sperm quality. The study aimed to establish a general pattern of lncRNA expression in each part of the turkey’s reproductive tracts. Since spermatogenesis in birds is very diverse, it should be helpful to characterize this process in the turkey. The introduced in silico approach indicated target genes and regulatory elements that should be further investigated to provide mechanistic insight. It would be very interesting to check how identified DEGs and DELs are reflected in the proteome and how this correlates with semen quality parameters. Further research including single-cell sequencing should also be considered to distinguish cell clusters within reproductive tracts and their role during spermatogenesis [80]. The results are a step forward and potentially may indicate directions for further research.

## 4. Materials and Methods

### 4.1. Tissue Collection

The turkeys came from an experimental farm of the Poultry Department (University of Warmia and Mazury in Olsztyn, Poland). As described previously [5], male turkeys were photostimulated at 26 wks of age (14 h light to 10 h darkness) and produced semen by 30 wks of age. Semen was collected at one-week intervals, by abdominal massage [67]. The sperm concentration, viability, motility and volume were 9.5 ± 1.5 × 10^9^ sperm/mL, 97.9 ± 0.3%, 85.2 ± 9.6% and 0.6 ± 0.05 mL, respectively. Male reproductive tract tissue samples were collected from six, 38-week-old turkeys that were slaughtered at a local slaughterhouse. Tissues were immediately frozen in liquid nitrogen and used for total RNA isolation. All experiments were conducted in accordance with the guidelines of the International Guiding Principles for Biomedical Research Involving Animals proposed by the Society for Reproductive Research and the Polish Act on Animal Welfare. These experiments did not require ethical consent. The sampling process was described in detail previously [5].

### 4.2. RNA Isolation, Library Preparation and Sequencing

All procedures from RNA isolation up to sequencing procedures were performed as described previously [81]. The key procedures in this process included the following steps. The total RNA was extracted using a Total RNA Mini kit, and the concentration was determined spectrometrically (NanoDrop-1000 Technologies LLC, Wilmington, DE, USA). The RNA integrity was checked using a 2100 Bioanalyzer, with an RNA 6000 Nano LabChip Kit and the samples with RIN above 8.0 and rRNA ratios above 1.0 were used for NGS. The final strand-specific RNA sequencing libraries were prepared by synthesizing the sequences with PCR and using the TruSeq Stranded mRNA Library Prep Kit (Illumina, San Diego, CA, USA) used according to the manufacturer’s protocol. The cDNA libraries were sequenced using the NovaSeq 6000 platform (Illumina, San Diego, CA, USA) as the paired-end, 2 × 100-bp sequences. Library preparation and RNA-Seq were outsourced to Macrogen (Macrogen, Inc., Seoul, Korea).

### 4.3. In Silico Preprocessing, Mapping and Expression Analyses

The raw reads, trimmed reads and mapping quality were checked using FastQC software, v. 0.11.7 (Bioinformatics Group at the Babraham Institute, Cambridge, UK; www.bioinformatics.babraham.ac.uk (accessed on 20 June 2022)). The trimming process was performed using the Trimmomatic software, v. 0.38 [82] and included adapter removing, low quality read filtering (Phred cutoff score ≤ 20; with 10-bp frameshifts) and reads trimmed to equal lengths = 90 nucleotides. Preprocessed reads were mapped to the turkey reference genome (version 2.01), using annotation version 2.01.91 and the transcriptome was then additionally annotated using genome version 5.01 with annotation version 5.01.104. The first reference was used because this research is a continuation of the analyses from the previous work [5], and it was crucial that the data could be compared directly and the 5.01.104 genome was used to analyze the release version with new lncRNA annotations.

The mapping was performed with the STAR software, v. 2.4 (Cold Spring Harbor Laboratory, Cold Spring Harbor, NY; https://github.com/alexdobin/STAR (accessed on 30 October 2021) [83] and StringTie, v. 1.3.3 (https://ccb.jhu.edu/software/stringtie (accessed on 8 October 2021)) [84] was used to enrich additional annotations by merging the reference GTF file with BAM files.

### 4.4. Expression Profiling and Differentially Expressed Genes

Expression profiles as the count matrices were enriched at both levels: transcripts and genes. The analysis of differential expression was performed on the gene level using the Ballgown [85] and Cuffdiff [86] methods. Genes were classified as the differentially expressed if the adjusted *p*-value was <0.01, the logarithmic value of expression fold-change was >1.0) and were confirmed by both methods. All statistical analyses were done in R Bioconductor packages [87].

### 4.5. The lncRNA Identification

The transcript sequences were extracted from the reference genome using the annotation file obtained in this research (merged gff files from all samples) with gffread script (https://github.com/gpertea/gffread (accessed on 30 October 2021)). The transcript count matrix was obtained simultaneously with the gene count matrix using merged GTF. It allowed control of the annotations on both gene and transcript levels. The reference turkey genome version 5.01.104 contains 1703 annotated lncRNA transcripts, and this genome was used (in the first step of the novel lncRNA identification process) to annotate (using BLAST similarity > 90%, E-value < 1 × 10^−10^) and extract known lncRNA transcripts from the transcriptomic dataset (Figure 1). The next steps of the filtering stage involved: removing all annotated protein coding transcripts, removing all transcripts of summarized expression lower than 10 counts, and filtering out all transcripts shorter than 200 nucleotides and consisted of only one exon. The filtered dataset was then analyzed for coding probability using seven methods: CPC2 [88], FEELnc [89], CPAT [90], PLEK [91], CNCI [92], LncFinder [93] and LncRNA Mdeep [94]. Only those sequences that were estimated as non-coding by at least five methods were finally filtered using the pFam [80] database and were classified as novel lncRNAs. Both datasets of novel and known lncRNAs were merged and used for the identification of differentially expressed lncRNAs (DELs) in each comparison. Consequently, lncRNAs were classified as DELs if they met the same conditions as DEGs (adjusted *p*-value < 0.01, log_2_(FC) > 1.0).

### 4.6. The lncRNA Interactions with Sperm Motility-Related DEGs

The merged three sets of DELs and DEGs were used to detect potential correlations and interactions. This work focused on the analysis of lncRNA relationships only with genes associated with sperm motility processes. The set of these genes is presented in Table 3 and has been described in detail in the authors’ previous manuscript [8]. These genes were classified into three groups based on their expression profiles. The first group DD/(T + E) consisted of processes enriched for the genes in which a significant difference of expression was detected between ductus deferens and other two tissues testis and epididymis. The group E/(T + DD) contains genes of differential expression in the epididymis in contrast to testis and ductus deferens, and the last group, T/(E + DD), contained the processes and genes significant in comparisons “testis/ductus deferens” and “testis/epididymis” (Figure 5). DEGs located on the genome closer than 100 kbp from DEL were classified as CIS-acting genes. TRANS-acting genes were those of similar expression profiles calculated as Pearson correlation and selected only those of r coefficient >0.9 and *p*-value < 0.05.

Direct interactions of DELs with potential target macromolecules were detected using two programs: LncTar v. 1.0 [95] and lncPro v. 1.0 [96] tools. LncTar calculated the binding strength of lncRNA to mRNA, and the results were considered significant when the ndG value was lower than –0.1 for each interaction. The lncPro program calculated the probability of hydrogen-bonding propensities and Van der Waal’s interaction between the secondary structures of the lncRNAs and proteins encoded by DEGs. If the probability was higher than 0.90, such interaction was considered significant.

The STRING [97] interaction network was constructed for proteins detected using the lncPro tool (lncRNA–protein interactions). The minimum required interaction score was set as 0.4, and all disconnected nodes were hidden. Most of the plots (such as Volcano and MA plots, Sankey plots of TRANS-acting relations, DELs heatmap) were prepared using the ggplot2 library in the R Bioconductor environment. The RNA–RNA interactions were visualized using the Cytoscape tool [98].

The whole process of novel lncRNA identification was performed with the homemade R library which is under preparation for publishing. All high-performance computations were performed using a 60-core CPU and a 136 GB RAM server at the Regional IT Center of the University of Warmia and Mazury in Olsztyn.

### 4.7. Real-Time PCR

As the validation of RNA-seq results, Real-time PCR was performed to detect the expression levels of *MSTRG.17922*, *MSTRG.6075*, *MSTRG.8989* and *MSTRG.25698*. Total RNA samples used in the NGS experiment were transcribed to cDNA using the QuantiTect^®^ Reverse Transcription Kit (Qiagen, Valencia, CA, USA), according to the manufacturer’s protocol. The expression of selected lncRNAs in testis, epididymis and ductus deferens were analyzed to validate obtained NGS data. Briefly, Real-time PCR reaction included the following: cDNA template, Power SybrGreen^®^ Master Mix (Life Technologies, Grand Island, NY, USA), RNase-free water to a final volume of 20 μL and primers selected for validation of NGS data. Each Real-time PCR reaction was performed in duplicate. Non-template controls were used for each set of primers to confirm the specificity of the reaction. The relative amplification of selected lncRNAs was calculated using the ΔΔCt method according to the Guide to Performing Relative Quantitation of Gene Expression Using Real-Time Quantitative PCR protocol and normalized using the geometric mean of the expression levels of a high stability reference gene in turkey testis, epididymis and ductus deferens, i.e., glyceraldehyde 3-phosphate dehydrogenase (*GAPDH*). Statistical analyses and graphs of the Real-time PCR results were made using the ‘pcr’ package [99] in the R environment. Statistical testing of PCR data was performed using the linear regression model [100].

## 5. Conclusions

We identified 863 new lncRNAs in the ductus deferens, testes and epididymis of turkeys and for the first time we characterize their potential role in the regulation of sperm maturation processes, in particular sperm motility. This offers the opportunity to further investigation of the function and role of lncRNAs in processes of molecular regulation.

## Figures and Tables

**Figure 1 ijms-23-07642-f001:**
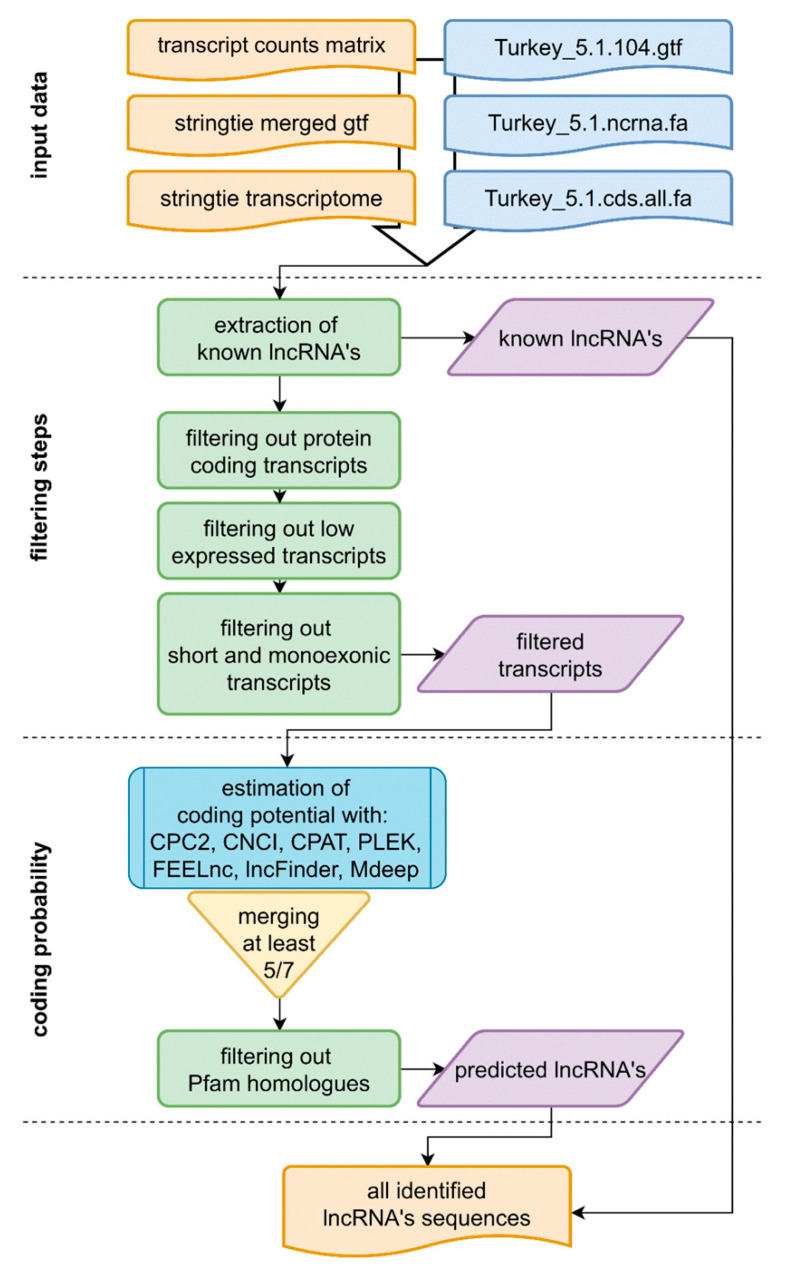
The flowchart of the lncRNA identification process. The process consists of several steps (described in the main text) grouped in the three stages: “input data”—preparing and reading the data from the previous steps (colored orange) and reference data (colored blue), “filtering steps”—filtering the transcripts basing on the annotation (i.e., extracting known lncRNAs and removing protein-coding transcripts) and structural features such as sequence length and the number of exons and “coding probability”—screening the sequence dataset based on the coding probability.

**Figure 2 ijms-23-07642-f002:**
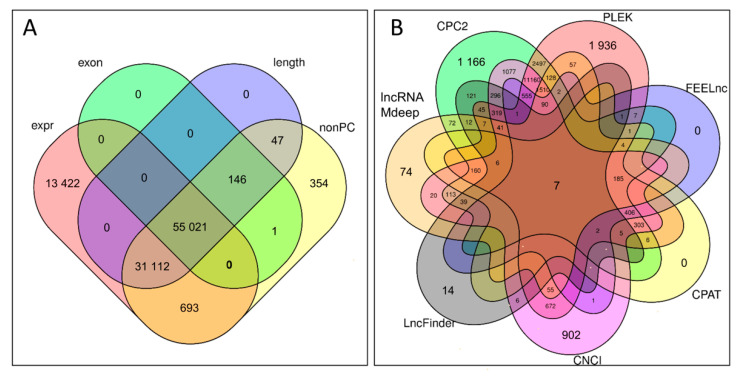
The Venn plots show the number of transcripts in the lncRNA identification process in the two stages: (**A**) filtering by basic parameters: expression level (expr), number of exons, sequence length, annotation to protein-coding genes (nonPC), (**B**) numbers of sequences predicted as non-coding using seven tools.

**Figure 3 ijms-23-07642-f003:**
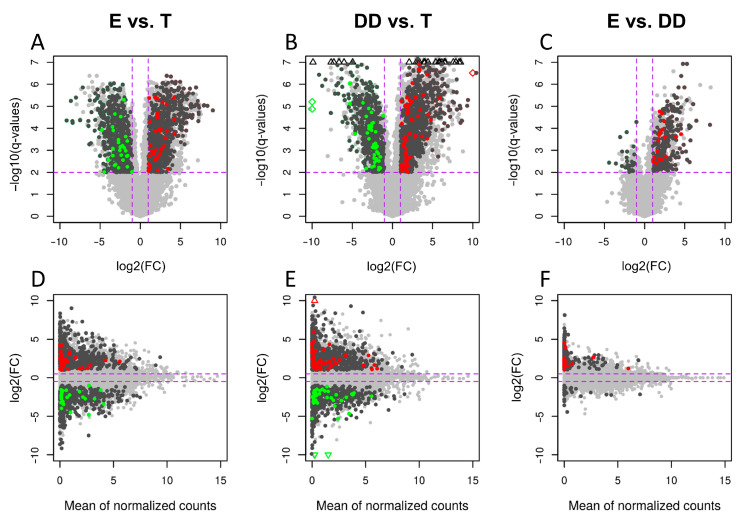
MA and volcano plots of expressed transcripts (presented on a gene level) in three comparisons: (**A**,**D**) epididymis vs. testis, (**B**,**E**) ductus deferens vs. testis and (**C**,**F**) epididymis vs. ductus deferens. Each dot presents expressed transcripts, where dark grey is DEGs, red is upregulated lncRNAs and green is downregulated lncRNAs. The triangles and rectangles indicate off the chart values.

**Figure 6 ijms-23-07642-f006:**
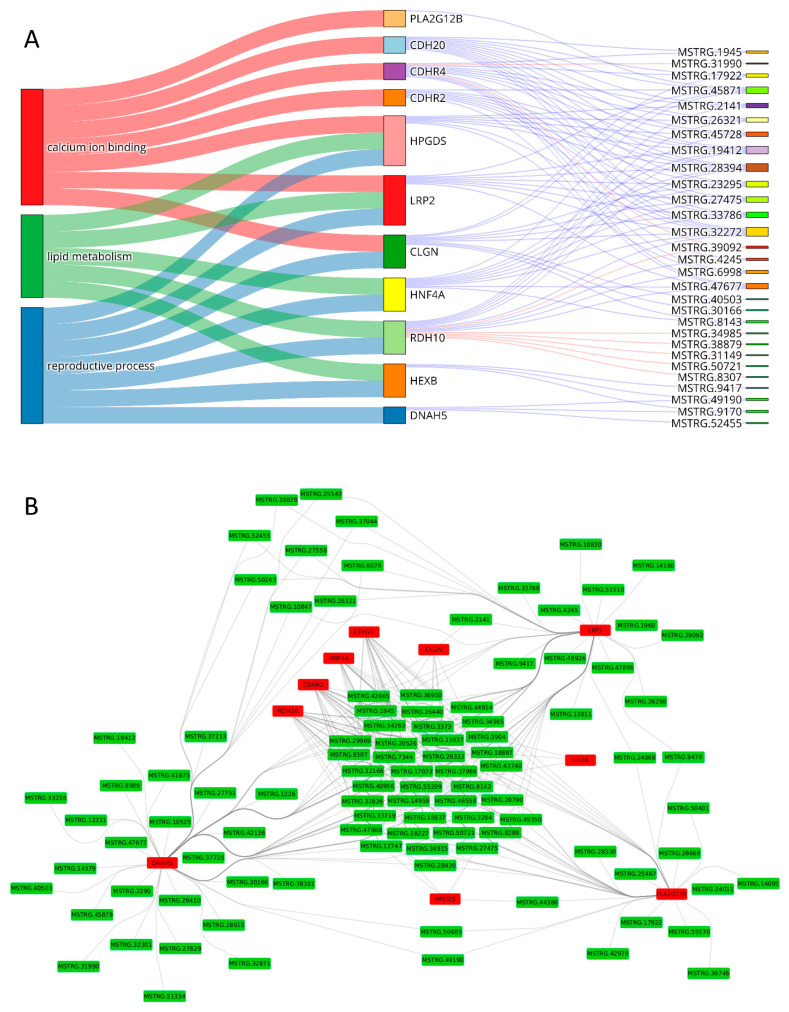
Trans-correlations (**A**) and direct lncRNA-mRNA interactions (**B**) identified in the group of processes with significantly expressed genes between DEGs and DELs associated with the E/(T + DD) group of molecular functions (see Table 4 and Figure 4). The first column in Figure A represents the biological processes. The second column shows the genes associated with the processes in the first column (the lines of the Sankey plot colored corresponding to the processes represent the gene–process relationship). The third column presents lncRNAs, and the lines connecting them with genes symbolize the expression correlation (blue—positive correlation, red—negative correlation). The green color in Figure B indicates lncRNAs, and red indicates the protein-coding genes. The lines symbolize the predicted direct interactions of the lncRNA-mRNA.

**Figure 7 ijms-23-07642-f007:**
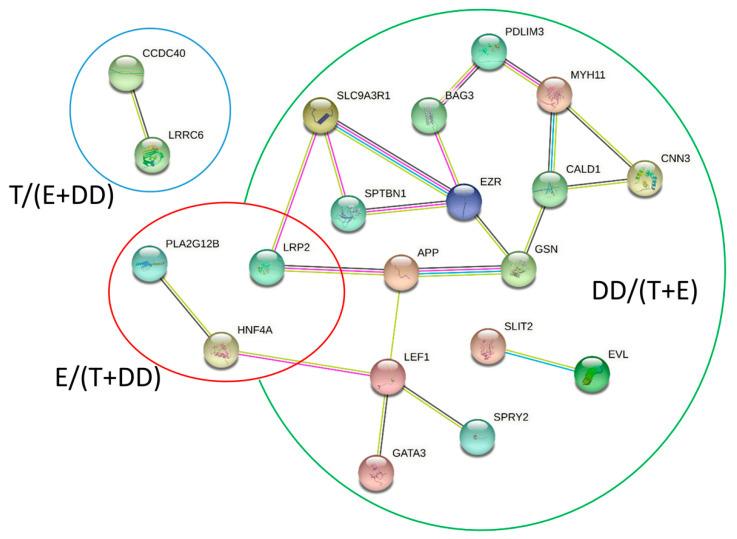
The STRING network of proteins involved in sperm motility. DD/(T + E), E/(T + DD) and T/(E + DD) groups correspond to the three groups of analyzed processes (see Table 4 and Figure 5). The color of the lines symbolizes the type of interaction and the source of information: light blue and pink-known interactions, green, red and blue-predicted interactions, yellow—text mining, black—co-expression, and purple—protein homology.

**Figure 8 ijms-23-07642-f008:**
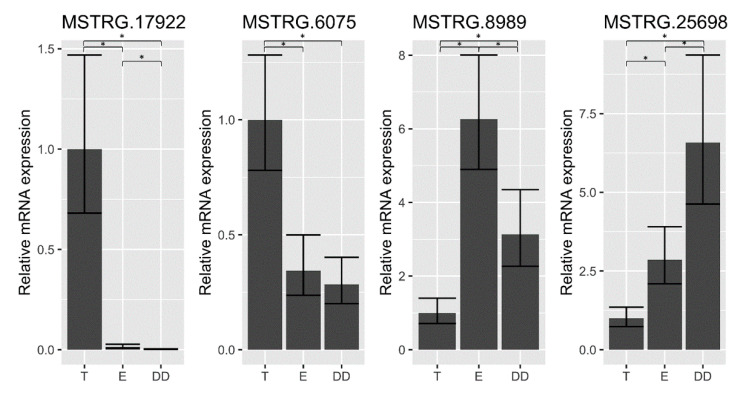
The real-time PCR results. The statistical significance of the expression differences was confirmed at the level <0.05 and is marked with a star. Data are shown as mean values ± SD.

**Figure 9 ijms-23-07642-f009:**
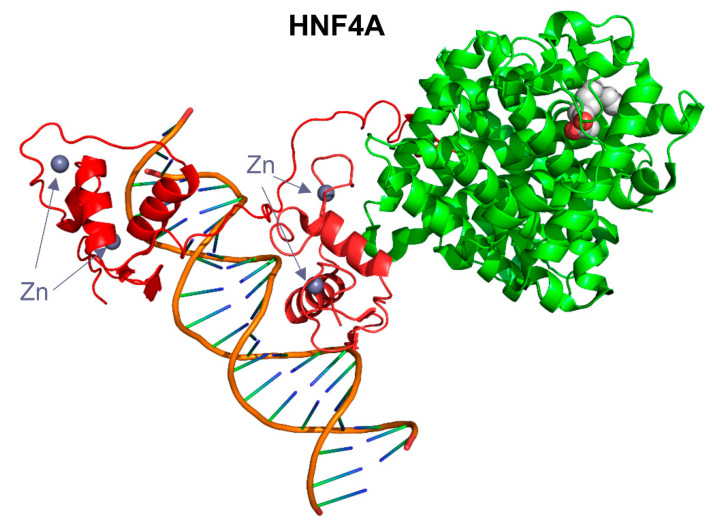
A visualization of the 3D model of the human HNF4A protein (PDB id 4IQR). The protein part is presented as ribbons, and the conserved DNA-binding domains are colored red.

**Table 1 ijms-23-07642-t001:** Overall statistics of sequenced, mapped and processed data. The read values are presented in millions. Min/max are the minimum and maximum values. Mean values are calculated for all six samples for each tissue.

	T		E		DD	
	Min/Max	Mean	Min/Max	Mean	Min/Max	Mean
**Raw reads**	127.9/165.8	146.9	126.4/176.9	152.0	102.6/167.7	145.9
**Trimmed reads**	110.7/140.6	123.8	96/151.9	127.5	86.2/137.6	121.4
**% of trimmed reads**	81.1/86.5	84.3	76/87.2	83.5	82/84.7	83.2
**Mapped reads**	78/100	87.6	64.8/117.3	93.0	62.3/99.9	84.8
**Uniquely mapped reads**	76.2/97.6	85.9	62.6/115.8	91.6	61.3/98.4	83.5
**% of uniquely mapped reads**	97.6/98.6	98.00	96.6/98.9	98.3	98.2/98.6	98.4
**Expressed transcripts**	59,151/73,569	67,594.3	22,109/61,868	47,862.8	33,355/47,861	41,412.5
**Expressed genes**	36,879/43,510	40,405.3	17,596/34,711	28,438.5	21,196/27,526	24,970.2

**Table 2 ijms-23-07642-t002:** The number of up- and down-regulated differentially expressed lncRNAs in three comparisons.

	E vs. T	DD vs. T	E vs. DD
All	139	182	25
Up-regulated	93	132	25
Down-regulated	46	50	0

**Table 3 ijms-23-07642-t003:** Differentially expressed genes (DEGs) associated with molecular processes engaged in sperm motility and grouped into three sets (T/(E + DD), E/(T + DD) and DD/(T + E) based on expression profiles presented in Figure 5.

Group	Process	Genes
T/(E + DD)	actin binding actin filament binding actin cytoskeleton organization actin filament-based process actin filament organization regulation of actin filament length regulation of protein phosphorylation apoptotic process response to oxygen-containing compound response to stress solute: cation symporter activity response to estrogen	*ARPC5, CALD1, CNN3, EVL, EZR, GSN, ITGB1, MICAL2, MYH11, PARVA, POF1B, SPTBN1, TWF1, USH1C, CARMIL1, CGNL1, EPHA1, FAT1, MET, NRP1, PACSIN2, PDLIM3, PHLDB2, SDCBP, SLC9A3R1, SLIT2, SORBS1, WDR1, ANXA1, IQGAP2, ACVR1, APP, BMP7, CTNNB1, HBEGF, NEDD9, PKD2, SPRY2, THBS1, ACKR3, BAG3, GATA3, LEF1, NET1, PTPRK, ATP1A1, CAT, DCN, MMP2, GCNT1, SOD, PRDX1, ATP6V0A4, PTPPK, FIGF, MYLK, SLC15A1, SLC34A2, KRT19*
E/(T + DD)	calcium ion binding reproductive process lipid metabolism	*CDHR2, CLGN, HPGDS, LRP2, CDHR4, PLA2G12B, CDH20, DNAH5, HEXB, HNF4A, RDH10*
DD/(T + E)	axonemal dynein complex assembly microtubule cytoskeleton organization microtubule-based movement inner dynein arm assembly outer dynein arm assembly flagellated sperm motility	*ARMC4, CCDC40, DNAH7, LRRC6, PIH1D3, ZMYND10, DEUP1, NDC80, PLK4, RSPH9, TTC26, CFAP221, ROPN1L, CFAP69*

**Table 4 ijms-23-07642-t004:** Cumulative correlations between DELs and DEGs. Significant correlations between lncRNA and mRNA or proteins in each comparison (number of interactions associated with the sperm motility/all identified interactions).

Relations/Interactions	Cis	Trans	lncRNA-mRNA	lncRNA-Protein
E vs. T	0/23	519/15,190	3302/10,171	5845/9712
DD vs. T	0/30	749/22,983	4875/13,938	7335/14,172
E vs. DD	0/1	99/759	221/317	405/512

## Data Availability

The whole project is available in the BioProject database in the NCBI under accession number: PRJNA597008. The raw data are deposited in the SRA database under SRP238360 ID. The expression tables and all transcriptomic sequences in fasta format are available in GEO database (GSE142428). All sequences of predicted lncRNAs were extracted from transcriptome fasta file and submitted to NCBI GenBank under accession numbers ranging from OM983514 to OM984417.

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
