# Peer review of "In Silico Identification of lncRNAs Regulating Sperm Motility in the Turkey (Meleagris gallopavo L.)"

_ijms, 2022, doi:10.3390/ijms23147642_

Round 1

Reviewer 1 Report

In this study, potential mechanisms of lncRNA regulation of processes related to sperm motility in turkey were investigated and described. Customized bioinformatics analysis was used to detect and identify lncRNAs, and their correlations with differentially expressed genes and proteins were also investigated. The fly in the ointment is that there are too few confirmatory tests, only RT-PCR.

Comment 1: The preface discusses the importance of spermatogenesis and spermatogenesis quality. Then why choose Turkey as the research object among birds? Suggest to be discussed.

Comment 2, lines 75: Why is "in silico" in italics?

Comment 3, lines 96: Were the six samples sequenced, six samples each from T, E and DD, or six samples in total?

Comment 4, lines 140: Pay attention to punctuation.

Comment 5, lines 146: I suggest "…A, B and C…" instead of "…A, B, and C…".

Comment 6: What do the green circle and the red triangle in figure 4 represent? It is suggested to add.

Comment 7, lines 191: Please unify the “Fig” or “Figure” in parentheses when referring to pictures in the text.

Comment 8, lines 235: Why are these lncRNAs selected for verification? What is its representativeness?

Comment 9, lines 248: It is recommended that all numbers above thousands be written in the same way, without commas or without commas.

Comment 10, lines 271: To italicize the gene, please check the full gene.

Comment 11, lines 355: Figure 9 shows the 3D model of human HNF4A protein. How much homology is there between human and Turkey HNF4A protein?

Comment 12, lines 357: As mentioned in this paper, many genes are found to be regulated by lncRNAs. Have you considered supplementary experiments to verify this?

Comment 13, lines 530: Note the indentation in the first line to keep the text consistent.

Comment 14, lines 618: Please check the reference format.

Comment 15: At the beginning, the paper divides into three groups: "E vs. T", "DD vs. T" and "E vs. DD", and then into three groups: "T/(E+DD)", "E/(T+DD)" and "DD/(T+E)". What is the significance and necessity of grouping? In addition, after the first occurrence of grouping abbreviations, it is suggested to use abbreviations in the following.

Reviewer 2 Report

In this manuscript, Jastrzebski et al., reported the identification and characterization of differentially expressed lncRNAs among testis, epididymis and ductus deferens of turkeys. The authors used qPCR to validate the sequencing results. They further used the in silico methods to analyze the ‘TRANS-acting genes’ and ‘RNA-RNA’ interactions. Overall, the experiments were nicely performed, the data analysis and presentation were done in a scientific fashion, and the figures and legends are largely clear. Given the scarcity of non-coding RNAs studies in the reproductive system in birds, and the fine quality of this manuscript, I recommend acceptance with a few minor changes / suggestions. 

Minor suggestions

1) L81-82 and table 1, please check ‘row reads’ or ‘raw reads’?

2) L464, please add info about ‘Marcrogen’ company in ‘()’.

3) L519, please provide reference for using ‘100 kbp’ as the criteria to identify CIS-acting genes.

4) In figure 3B and E, what do those small triangles and rectangles mean?
